# Sex-specific role for the long noncoding RNA *Pnky* in mouse behavior

Parna Saha[1,2], Rebecca E. Andersen [1,2,6], Sung Jun Hong[1,2], Eugene Gil[1,2], Jeffrey Simms[3], Hyeonseok Choi[4] & Daniel A. Lim [1,2,5] ✉

The aberrant expression of specific long noncoding RNAs (lncRNAs) has been associated with cognitive and psychiatric disorders. Although a growing number of lncRNAs are now known to regulate neural cell development and function, relatively few lncRNAs have been shown to underlie animal behavior. *Pnky* is an evolutionarily conserved, neural lncRNA that regulates brain development. Using mouse genetic strategies, we show that *Pnky* has sex-specific roles in mouse behavior and that this lncRNA can underlie specific behavior by functioning in *trans*. Male *Pnky*-knockout mice have decreased context generalization in a paradigm of associative fear learning and memory. In female *Pnky*-knockout mice, there is an increase in the acoustic startle response, a behavior that is altered in affective disorders. Remarkably, expression of *Pnky* from a bacterial artificial chromosome transgene decreases the acoustic startle response in female *Pnky*-knockout mice, demonstrating that *Pnky* can modulate specific animal behavior by functioning in *trans*. More broadly, these studies illustrate how specific lncRNAs can underlie cognitive and mood disorders.

Long noncoding RNAs (lncRNAs), transcripts longer than 200 nucleotides (nts) that do not encode proteins, have emerged as key regulators of important biological processes[1,2]. Of the tens of thousands of distinct lncRNAs produced by the mammalian genome, many are brain and neural cell type specific[3–7], and aberrant lncRNA expression has been implicated in a broad range of neurological disorders, including schizophrenia, depression, autism, and Alzheimer's disease[8,9]. However, although the list of lncRNAs experimentally shown to regulate neural cell biology is expanding[10–12], very few lncRNAs have been demonstrated to have roles in animal behavior[13–20].

*Pnky* is an evolutionarily conserved, neural lncRNA that regulates mouse neural stem cell (NSC) function both in vitro and in vivo[21,22]. *Pnky* conditional knockout (cKO)[22] or transcript knockdown (KD)[21] increases neuronal production from cultured mouse NSCs, and other studies indicate a role for *Pnky* in NSC migration[23]. In vivo, genetic deletion of *Pnky*−either cKO or germline KO−causes aberrant cortical development, resulting in alterations of neuron subtype abundance[22].

Whether *Pnky*-KO would have an effect upon behavior was very unclear. Only a small number of lncRNAs have been genetically disrupted in mice, and in certain well-studied cases, behavior is only mildly affected[13] or apparently not at all[24]. Although the cellular phenotype of acute *Pnky*-deletion in cultured NSCs is dramatic (e.g., ~4-fold increase in neuronal differentiation), *Pnky*-deletion in vivo produces relatively subtle changes in cortical anatomy (e.g., ~15% difference in specific layers of the cortex), suggesting that the in vivo context provides compensatory molecular mechanisms for the absence of *Pnky* in neurodevelopment[21,22]. Given that animal behavior includes many additional layers of potential compensatory mechanisms (e.g., the plasticity of neuronal circuits), observing a behavioral

[1]Department of Neurological Surgery, University of California, San Francisco, San Francisco, CA 94143, USA. [2]Eli and Edythe Broad Center of Regeneration Medicine and Stem Cell Research, University of California, San Francisco, San Francisco, CA 94143, USA. [3]Behavioral Core, Gladstone Institutes, San Francisco, CA 94158, USA. [4]Department of Molecular and Cell Biology Undergraduate Program, University of California, Berkeley, Berkeley, CA 94720, USA. [5]San Francisco Veterans Affairs Medical Center, University of California, San Francisco, San Francisco, CA 94143, USA. [6]Present address: Division of Genetics and Genomics, Harvard Medical School, Boston Children's Hospital, Boston, MA 02115, USA. ✉e-mail: Daniel.Lim@ucsf.edu

phenotype in a neural lncRNA-KO mouse is a relatively demanding benchmark to establish the importance of that lncRNA.

Gene-disrupting manipulations to the genome can disrupt potential *cis* function of the locus. For the study of lncRNA function, this consideration is particularly important since many lncRNAs appear to function in *cis*, regulating the expression of neighboring genes[25,26]. Even the use of KD approaches (e.g., short-hairpin RNAs and antisense oligonucleotides) does not distinguish *cis* from *trans* function, since transcript KD can disrupt transcriptional elongation, which itself modulates neighboring gene expression[27]. In previous studies, neither *Pnky*-deletion nor *Pnky* KD alters the expression level of any other gene within a 1 MB chromosome window, suggesting that *Pnky* does not function in *cis*[21,22].

LncRNA expression from a bacterial artificial chromosome (BAC) transgene can be used to test for *trans* function and further demonstrate function of the lncRNA itself[25,27]. For example, genetic disruption of lncRNA *Fendrr* causes defects in heart development, and expression of *Fendrr* from a BAC transgene rescues some of the developmental abnormalities, indicating that *Fendrr* can function in *trans*[28]. In a previous study[22], we generated a transgenic mouse line that expresses *Pnky* from a ~170 kb BAC insert (BAC-*Pnky*) that lacks other known genes, and BAC-*Pnky* rescues transcriptomic, cell culture, and in vivo developmental phenotypes of *Pnky*-deletion. These results strongly support the notion that *Pnky* encodes a *trans*-acting functional lncRNA. However, whether the *trans*-action of *Pnky* can also underlie the much more complex phenotypes of animal behavior was unknown, and *trans* rescue of a lncRNA-KO behavioral phenotype remains a highly stringent benchmark—one that to the best of our knowledge has not yet been achieved—to demonstrate lncRNA function.

In this study, we investigated the role of *Pnky* in adult mouse behavior with *Pnky*-KO and BAC-*Pnky* mice. We discovered sex-specific phenotypes of *Pnky*-KO in animal behaviors that relate to cognition and certain affective disorders. Furthermore, we found that BAC-*Pnky* can selectively reverse an important *Pnky*-KO behavioral phenotype, also in a sex-specific manner. In addition to characterizing *Pnky* as a lncRNA important to animal behavior, these data more broadly illustrate how lncRNAs can underlie cognitive and psychiatric disorders.

## Results

### *Pnky*-KO does not disrupt basic neurological function and nesting behavior

*Pnky*-KO mice are born at expected Mendelian frequencies and do not have obvious major deficits of general health and neurological behavior[22]. Across multiple brain regions, including the cortex, hippocampus and amygdala, reverse transcription-quantitative polymerase chain reaction (RT-qPCR) analysis confirmed the absence of *Pnky* transcripts in the *Pnky*-KO animals (Supplementary Fig. 1). To more systematically assess behavior, we analyzed a cohort of *Pnky*-KO (*n* = 12 female (XX) and 14 male (XY)) mice and littermate *Pnky* +/+ (WT, *n* = 11 XX and 11 XY) controls with a panel of behavioral tests 3–5 months after their birth (Fig. 1a). Data from behavioral tests were collected by experimenters blinded to the genotype.

To assess locomotor function, we performed the open field test (OFT)[29] in which ambulation within and across different zones in an open, wall-enclosed area is measured using photobeam arrays. The amount of total movement, ambulatory movement and rearing behavior was not different between *Pnky*-KO mice and controls (Fig. 1b, Supplementary Fig. 2a,b), indicating normal levels of spontaneous motor activity across the different experimental groups.

To evaluate motor coordination and balance, we employed the rotarod test[30]. Neither XX nor XY *Pnky*-KO mice exhibited differences in their latency to fall in the training (constant speed, 16 rotations per minute (RPM)) or the testing (accelerating, 4-40RPM) phases as compared to their WT controls (Fig. 1c).

Nest building is an important task for rodents that requires complex motor skills and spatial memory. Impaired nest building is also a sensitive indicator of decreased general health and sense of well-being[31]. In both XX and XY mouse cohorts, the nesting score of *Pnky*-KO mice was not different from that of their WT littermates (Fig. 1d). Thus, *Pnky* is not required for general locomotor function, motor coordination, balance, and the complex behavior of nest building.

### *Pnky*-KO mice do not exhibit changes in anxiety or deficits in social interactions

In OFT, the fraction of time spent in the open center of the field versus that of the walled perimeter is a measure of anxiety, and there was no difference with *Pnky*-KO (Supplementary Fig. 3a). To further investigate anxiety-related behaviors, we used the elevated plus maze (EPM) test[32]. The EPM is a raised platform in the shape of a plus sign (+), consisting of two open arms (no walls) that are intersected by two enclosed arms (with walls). Mice with increased anxiety avoid the elevated, open spaces of the open arms and spend more time in the enclosed arms of the EPM. *Pnky*-KO mice were not different from controls in EPM testing (Fig. 2a, Supplementary Fig. 3b), further indicating a normal, baseline level of anxiety in *Pnky*-KO mice.

The two-chamber social approach test measures the preference of mice for interactions with other mice[33]. For this test, mice were allowed to explore an arena that has two chambers, one of which contains a wire enclosure with an inanimate mouse toy (inanimate chamber) and the other containing a wire enclosure with a live mouse (social chamber). Mice with social deficits generally have decreased interaction bouts and time with the stimulus mouse as compared to the inanimate mouse. Both sexes of the *Pnky*-KO and WT mice interacted significantly more with the social chamber, and there were no differences between the two genotypes (Fig. 2b), demonstrating a normal level of social interaction in *Pnky*-KO mice.

### XY *Pnky*-KO mice have decreased fear context generalization

To investigate cognitive behavior, we employed two different testing paradigms that evaluate different aspects of learning and memory. The object-context congruence test measures visual recognition of objects and their association with different environmental contexts[34]. After training to associate specific objects (blocks vs. flasks) with different environmental contexts (white walls vs. checkered pattern walls), mice were tested for whether they can distinguish "congruent" object-context associations (e.g., block object with white wall context) from "incongruent" associations (e.g., block object with checkered pattern wall context). *Pnky*-KO and WT mice did not exhibit significant differences in either the training or test phases of the object-context congruence test (Supplementary Fig. 4a,b).

The cued fear conditioning and recall test measures the ability of mice to learn and remember an association between environmental cues and aversive experiences[35]. This test of Pavlovian conditioning consists of three phases: (1) cued fear training, (2) fear context testing and (3) cued fear recall testing. During the training phase, a tone is paired with a mild electric shock, and mouse freezing behavior is quantified by video analysis. In normal mice, with successive presentations of tone-shock pairs, the amount of freezing increases, which corresponds to associative learning. Reactivity to the foot-shocks given during the cued fear conditioning test as measured by the motion index and latency time for hind-paw withdrawal to hot-plate stimulation were not different between *Pnky*-WT and *Pnky*-KO animals, regardless of sex (Supplementary Fig. 5a,b). In both sexes, during cued fear training, the freezing behavior of *Pnky*-KO mice was not significantly different from their WT controls (Fig. 3a, Supplementary Fig. 6a), indicating a similar degree of learning of this behavior among *Pnky*-KO and WT mice.

Fear context testing is performed 24 h later in the same environmental context (same box as training phase) where freezing behavior is

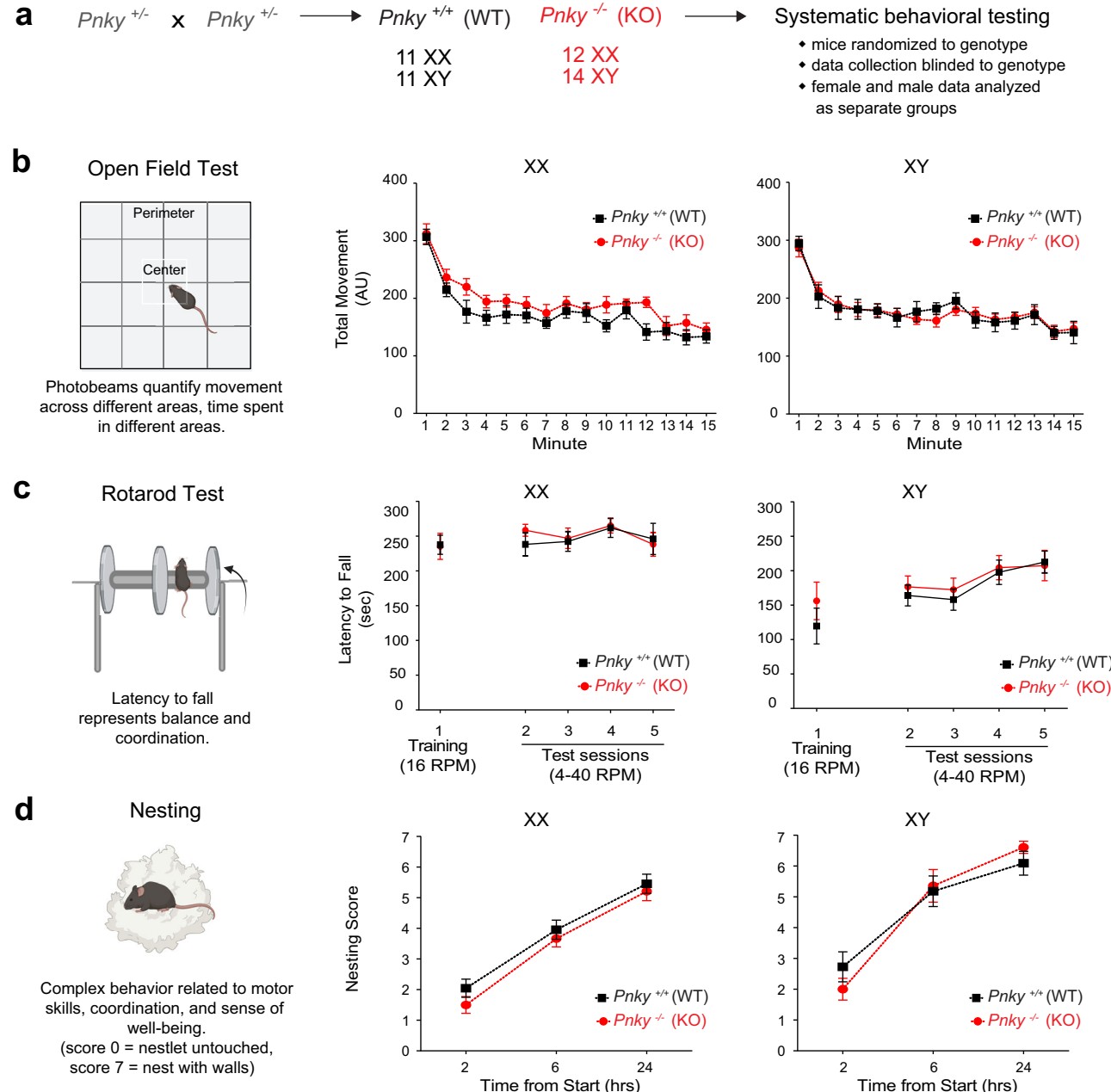

**Fig. 1 | *Pnky*-KO does not result in basic neurological deficits. a** *Pnky* +/− mice were crossed to obtain a cohort of *Pnky*-KO (*n* = 12 XX and 14 XY) mice and littermate *Pnky* +/+ (WT, *n* = 11 XX and 11 XY). **b** Open field test assesses the gross activity levels (locomotion) and natural exploration habits in rodents. Total movements (measured in arbitrary units (AU)) in the open field (center and outer zones) are comparable between *Pnky*-WT and *Pnky*-KO animals; repeated measures two-way ANOVAs, *p* = ns (XX- *p* = 0.1075, XY- *p* = 0.9990). **c** Rotarod test measures balance and motor coordination. Latency to fall on the Rotarod test in the training; two-tailed, unpaired Welch's *t* tests, *p* = ns (XX: *p* = 0.9218, XY: *p* = 0.3429) and test phases; repeated measures two-way ANOVAs, *p* = ns (XX: *p* = 0.8284, XY: *p* = 0.5726) shows no difference between the two genotypes. Rotarod mean session data is reported, each session is comprised of 3 trials separated by a 15–20 min ITI. **d** Nest building is an innate behavior to assess general well-being. Nesting behavior is unaffected by *Pnky* deletion across the 24 hr testing period; repeated measures two-way ANOVAs, *p* = ns (XX: *p* = 0.2708, XY: *p* = 0.9791). ns = non-significant, data is represented as mean ± SEM. Source data are provided as a Source Data file. Cartoons in this figure are created with Biorender.com.

again quantified. This second phase evaluates whether mice have learned to associate the environmental context with the aversive experience (foot shock). To get a pure measure of contextual memory, no auditory cues or foot shocks are presented during this phase. During fear context testing, the amount of freezing behavior was not different between *Pnky*-KO and WT mice, regardless of sex (Fig. 3b), indicating that *Pnky*-KO does not impair memory of the fear-context association.

The third phase−cued fear recall testing−occurs 24 h after fear context testing. In this test, the visual and tactile appearance of the box is changed, which removes the prior environmental context associations. The same repetitive pattern of cues (tones) is presented but without any foot shocks, and mouse freezing behavior is quantified. In XX *Pnky*-KO mice, the amount of freezing with the auditory cues was not different from that of WT controls (Fig. 3c). In contrast, XY *Pnky*-KO mice exhibited decreased freezing behavior in the 5-min context

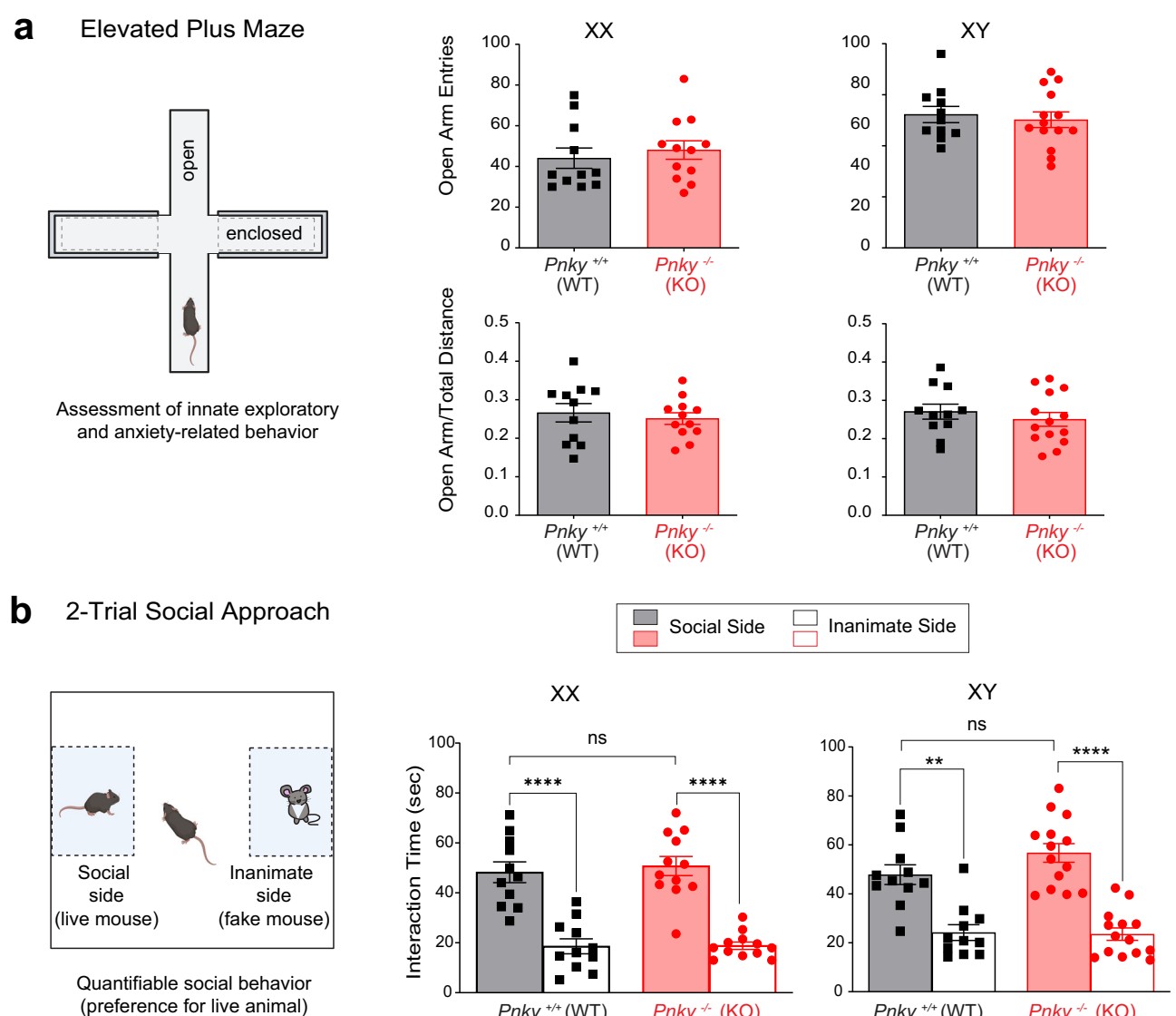

**Fig. 2 | *Pnky*-KO mice do not exhibit increased anxiety or deficits in social interactions. a** The elevated plus maze (EPM) test is used to assess anxiety related behavior. In the EPM, the number of entries into the open arm (XX, Mann-Whitney test, $p$ = ns (0.3705); XY, Welch's $t$ test, $p$ = ns (0.6448)) and the ratio of open arm distance to total distance (Welch's $t$ tests, $p$ = ns (XX: $p$ = 0.6002, XY: $p$ = 0.4564) was equal for both the genotypes. **b** The social approach test assesses general sociability and preference for social novelty. In the social approach trial, XY and XX animals of both genotypes show increased preference (interaction time in seconds) for the social side, whereas the difference in interaction time across genotypes is non-significant. Paired $t$ test ****$p$ < 0.0001 for social vs inanimate for both

*Pnky*-WT and *Pnky*- KO XX animals; XX *Pnky*-WT vs *Pnky*-KO social interaction, Welch's $t$ test, $p$ = ns (0.6648). Paired $t$ test **$p$ = 0.0029 for social vs inanimate for *Pnky*-WT XY mice and paired $t$ test ****$p$ < 0.0001 social vs inanimate for *Pnky*-KO XY mice; XY *Pnky*-WT vs *Pnky*-KO social interaction, Welch's $t$ test, $p$ = ns (0.1228). **$p$ < 0.01, ****$p$ < 0.0001, ns = non-significant, data is represented as mean ± SEM. Unpaired Welch's $t$ test, Mann-Whitney test, and paired $t$ tests used are two-tailed. For EPM and Social approach trial test, $n$ = 11 XX and 11 XY *Pnky*-WT and $n$ = 12 XX and 14 XY *Pnky*-KO mice. Source data are provided as a Source Data file. Cartoons in this figure are created with Biorender.com.

generalization phase (48.2% decrease, *$p$ = 0.0101), during the tone presentations (18.8% decrease *$p$ = 0.0183), and in the inter-cue intervals (ICI), (23.46% decrease *$p$ = 0.0152) (Fig. 3c, Supplementary Fig. 6b). Given that the freezing behavior was reduced during context generalization in XY *Pnky*-KO mice, and differences during tone presentation and the ICI could be attributed to this (Supplementary Fig. 6c), the most conservative interpretation of these data is that there is decreased context generalization in XY *Pnky*-KO mice. Taken together, these studies indicate that *Pnky* is required for a specific type of associative memory in a sex-specific manner.

### *Pnky*-KO increases the acoustic startle response in XX mice

The acoustic startle response (ASR) in mice is a non-stereotypic behavior that is characterized by a quantifiable flinching behavior in response to a loud (100–120 dB) sound stimulus. Differences in baseline startle responses can occur with various psychiatric conditions and specific brain lesions[36–38]. To measure ASR, mice are placed on a movement sensor plate (that measures the amplitude of the startle, flinching behavior) and exposed to a series of tones of varying intensities at random intervals for 15 min. XX *Pnky*-KO mice exhibited greatly increased startle amplitudes in response to tones across a broad range (80 to 120 dB) of acoustic intensities (0 dB - ns, 70 dB – ns, 80 dB -**$p$ = 0.0045, 90 dB - ***$p$ = 0.0009, 100 dB - **$p$ = 0.0045, 110 dB - *, $p$ = 0.0439, 120 dB - *$p$ = 0.0439) with 55.4% increase at 120 dB compared to the *Pnky*-WT XX mice (Fig. 4a). In XY *Pnky*-KO mice, the startle amplitude at each tone intensity was not different from that of their WT controls (Fig. 4a). Thus, loss of *Pnky* greatly increases the ASR of XX mice.

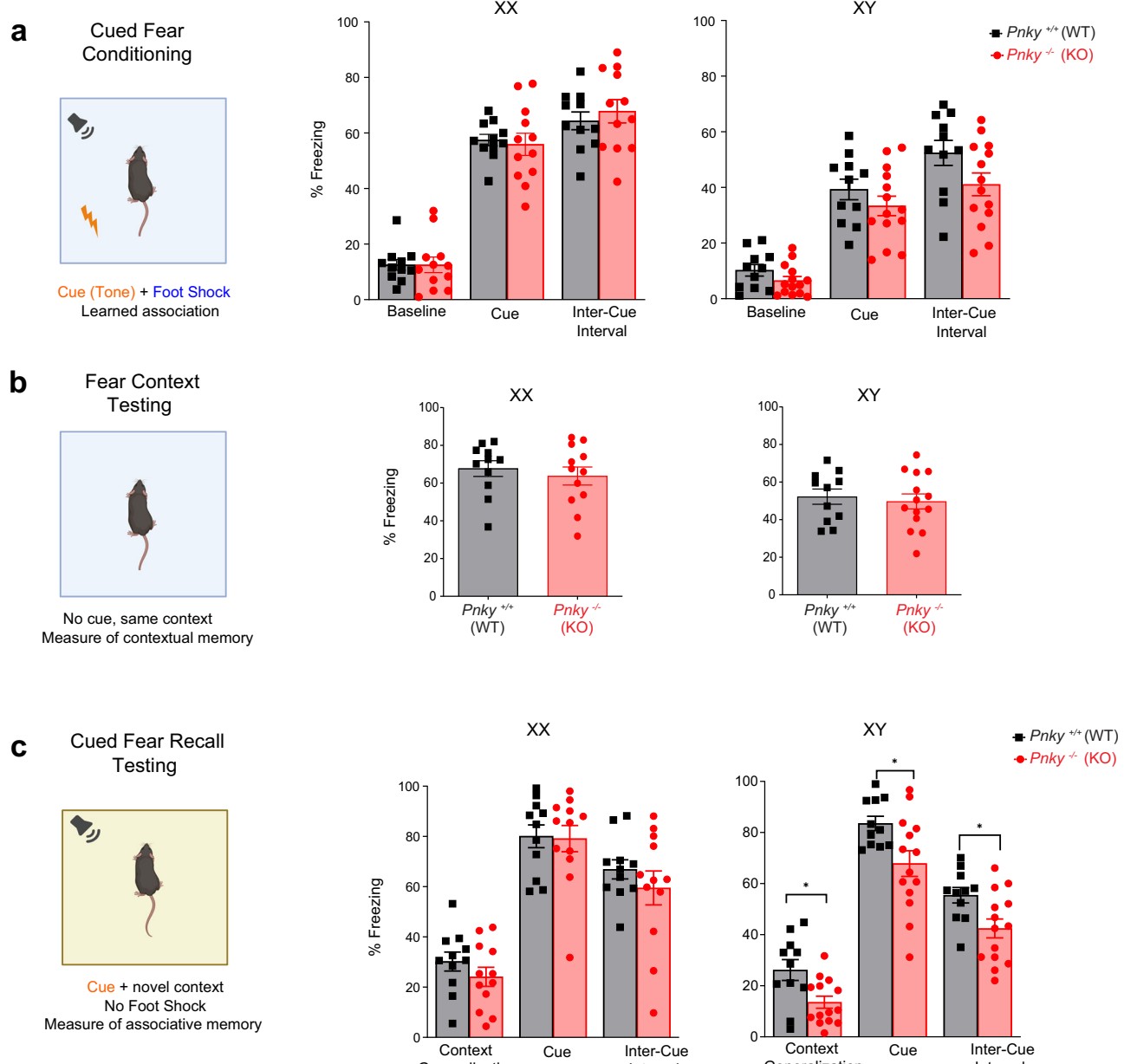

**Fig. 3 | XY *Pnky*-KO mice have decreased fear context generalization.** Cued fear conditioning test assesses the ability of mice to learn and remember an association between environmental cues and aversive experiences measured by recording the freezing response. % Freezing (lack of movement aside from normal respiration) is calculated in 1 min blocks. **a** During cued fear conditioning freezing behavior in *Pnky*-KO mice was comparable to its WT controls (multiple unpaired *t* tests (Holm-Šídák method) for the baseline, cue presentations and ICI for XX and XY mice, *p* = ns (Baseline XX: *p* = 0.9980, XY: *p* = 0.1463; Cued XX: *p* = 0.7454, XY: *p* = 0.2636; ICI XX: *p* = 0.5250, XY: *p* = 0.0761). **b** In the fear context test, the freezing behavior show no difference between the *Pnky*-WT and *Pnky*-KO groups; two-tailed unpaired Welch's *t* tests, *p* = ns (XX- *p* = 0.5404, XY- *p* = 0.6604). **c** During cued fear recall test, *Pnky*-WT and *Pnky*-KO XX animals exhibit no significant difference in freezing behavior (multiple *t* tests (Holm-Šídák method) for generalization (*p* = 0.2690), cue presentations (*p* = 0.8970) and ICI (*p* = 0.3631). *Pnky*-KO XY mice show decreased freezing compared to the WT control in the 5-min generalization period multiple *t* tests for the generalization (**p* = 0.0101), cue presentations (**p* = 0.0183), and ICI, (**p* = 0.0152). **p* < 0.05, ns = non-significant, data is represented as mean ± SEM. For the cued fear conditioning and recall tests, *n* = 11 XX and 11 XY *Pnky*-WT and *n* = 12 XX and 14 XY *Pnky*-KO mice. Source data are provided as a Source Data file. Cartoons in this figure are created with Biorender.com.

Prepulse inhibition (PPI) of ASR is the reduction in the amplitude of the startle response when a weak, non-startling tone – the prepulse – precedes the loud, startling tone. PPI is considered to represent a type of sensorimotor gating, wherein sensory information is "filtered" in the brain before a motor or cognitive response[39]. To measure PPI, mice are exposed to a series of acoustic startle tones (120 dB) in which some are preceded by weaker prepulse tones of different intensities (4, 15, and 26 bB) above background noise. Startle amplitude is quantified, and for each prepulse intensity, PPI is represented as the percent

reduction of the startle amplitude as compared to the baseline (non-prepulse) startle amplitude. In XY *Pnky*-KO mice, PPI across the different prepulse intensities was not different from that of their WT controls (Fig. 4b). In contrast, at the lowest prepulse intensity (4 dB above background) XX *Pnky*-KO mice exhibited reduced PPI (***p* = 0.0068), (Fig. 4b). Given that PPI was reduced only at low prepulse intensity, and that a high ASR can contribute to that observation, the most conservative interpretation of these results is that *Pnky*-KO increases the ASR in XX mice.

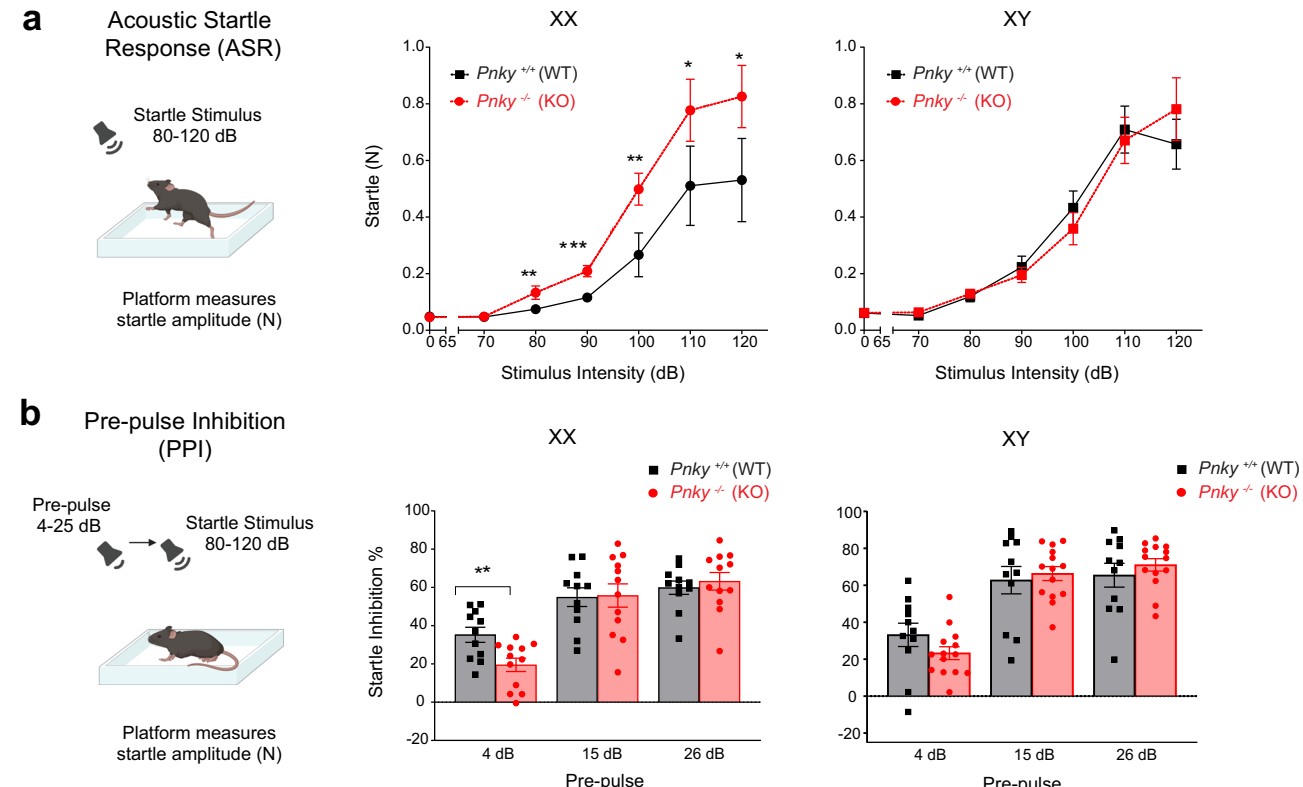

**Fig. 4 | *Pnky*-KO increases the acoustic startle response in XX mice. a** Acoustic startle response (ASR) measures quantifiable flinching behavior in response to a loud (80–120 dB) auditory stimulus. *Pnky*-KO XX mice exhibit increased startle with increasing intensity of the acoustic stimuli (rank summary analysis overall effect **$p = 0.0022$ across the whole range of stimuli, followed by Welch's *t* tests at 0 dB: $p = 0.5760$, 70 dB: $p = 0.6911$, 90 dB: ***$p = 0.0009$ and Mann-Whitney test at 80 dB: **$p = 0.0045$, 100 dB: **$p = 0.0045$, 110 dB: *$p = 0.0439$, 120 dB: *$p = 0.0439$). Startle response in *Pnky*-KO XY animals was unchanged (rank summary analysis overall effect $p = 0.8915$). **b** Prepulse inhibition (PPI) of ASR is the reduction in startle response when a weak, non-startling tone (prepulse) precedes the auditory startle stimulus. *Pnky*-KO XX mice exhibit decreased PPI at the prepulse intensity,

4 dB over the background: **$p = 0.0068$, multiple unpaired *t* tests (Holm-Šídák method) but not at 15 dB ($p = 0.9159$) and 26 dB ($p = 0.5739$) above the background. PPI was not altered in XY *Pnky*-KO mice; multiple unpaired *t*-tests (Holm-Šídák method), $p =$ ns for all prepulse intensities 4 dB ($p = 0.1614$), 15 dB ($p = 0.6516$), and 26 dB ($p = 0.4216$) above the background. *$p < 0.05$, **$p < 0.01$, ***$p < 0.001$, ns = non-significant, data is represented as mean ± SEM. For ASR and PPI tests, n = 11 XX and 11 XY *Pnky*-WT and n = 12 XX and 14 XY *Pnky*-KO mice. Unpaired Welch's *t* test and Mann-Whitney test used are two-tailed. Source data are provided as a Source Data file. Cartoons in this figure are created with Biorender.com.

## BAC-*Pnky* reduces the acoustic startle response of XX *Pnky*-KO mice

BAC-*Pnky* rescues transcriptomic, cellular and neurodevelopmental phenotypes of *Pnky* deletion[22]. However, whether a lncRNA expressed in *trans* can reverse behavioral phenotypes—which are fundamentally more complex than molecular, cellular and even developmental phenotypes—has been unclear. We therefore generated a cohort of *Pnky*-KO (15 XX and 18 XY) and *Pnky*-KO;BAC-*Pnky* (13 XX and 16 XY) littermates for behavioral testing during the 4th to 6th month after birth (Fig. 5a).

General neurological function as assessed by OFT was not different between *Pnky*-KO and *Pnky*-KO;BAC-*Pnky* mice in either sex (Supplementary Fig. 7a). Assessment of anxiety and social interactions with the EPM and two-chamber social approach tests, respectively, also did not reveal behavioral differences between *Pnky*-KO and *Pnky*-KO;BAC-*Pnky* mice (Supplementary Fig. 8a,b). Thus, BAC-*Pnky* does not produce behavioral changes in *Pnky*-KO mice among behaviors that were not different in *Pnky*-KO mice versus the WT controls.

XY *Pnky*-KO mice displayed a significant decrease in fear context generalization (Fig. 3c), but BAC-*Pnky* did not change measures of this type of associative fear learning and memory mice in either XX or XY genotypes (Supplementary Fig. 9a–c). Reactivity to the foot-shocks given during the cued fear conditioning test as measured by the motion index and latency time for hind-paw withdrawal to hot-plate

stimulation were not different between *Pnky*-KO and *Pnky*-KO;BAC-*Pnky* animals, regardless of sex (Supplementary Fig. 10a,b), indicating that the lack of a BAC-*Pnky* phenotype in this cohort is not due to differences in shock reactivity or pain response. Similarly, while *Pnky*-KO XX mice exhibited lower PPI as compared to their WT controls, BAC-*Pnky* did not increase PPI in either sex as compared to the *Pnky*-KO littermates (Supplementary Fig. 11). These data indicate that BAC-*Pnky* cannot reverse certain behavioral phenotypes related to *Pnky*-KO.

When BAC-*Pnky* was present, XX *Pnky*-KO mice exhibited a strong decrease in ASR (Fig. 5b). In contrast, in XY *Pnky*-KO mice, the presence of BAC-*Pnky* did not change the ASR (Fig. 5b). The amount of ASR reduction (41.5%) observed with BAC-*Pnky* in *Pnky*-KO XX mice is analogous to the amount of ASR increase (55.4%) observed with *Pnky*-KO vs. WT (Fig. 5c), suggesting that BAC-*Pnky* produces an opposite effect on ASR phenotype when compared to its corresponding *Pnky*-KO XX mice cohort.

In a set of XX and XY animals from these two cohorts, we used RNA fluorescent in situ hybridization (FISH) to assess *Pnky* expression in the cortex, hippocampus and amygdala (Supplementary Fig. 12a,c,e). As expected, similar to previous studies, *Pnky* expression was deficient in *Pnky*-KO brain regions, and *Pnky* levels in *Pnky*-KO;BAC-*Pnky* mice were 32–55% of control across the different brain regions (Supplementary Fig. 12b,d,f). Expression of *Pnky* from the BAC transgene was detected in both XY and XX mice at similar levels, thus,

**a**

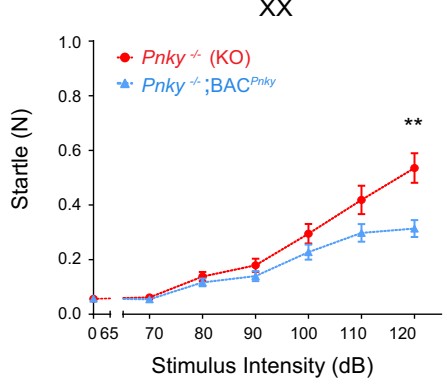

$Pnky^{-/-}$ x $Pnky^{-/-}$; BAC$^{Pnky}$ → $Pnky^{-/-}$ (KO)    $Pnky^{-/-}$; BAC$^{Pnky}$ → Behavioral testing

15 XX    13 XX
18 XY    16 XY

- general neurological function (OFT, EPM)
- 2-chamber social approach
- fear conditioning and recall
- ASR, PPI

**b**   Acoustic Startle Response (ASR)

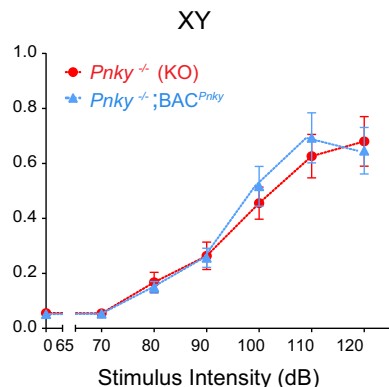

**c**   ASR at 120 dB *Pnky*-WT *vs Pnky*-KO and *Pnky*-KO *vs Pnky*-KO; BAC-*Pnky*

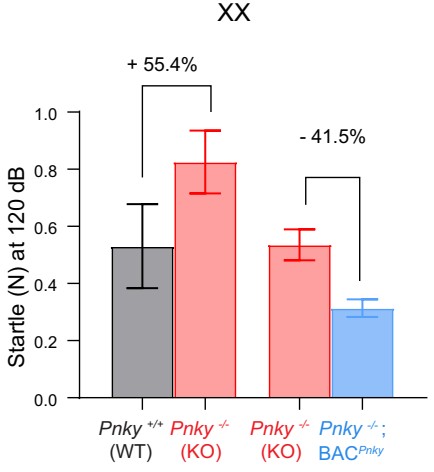

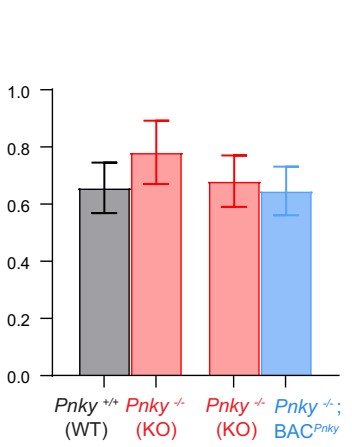

**Fig. 5 | BAC-*Pnky* reduces the acoustic startle response of XX *Pnky*-KO mice.** **a** *Pnky*−/− mice were crossed to *Pnky*−/−; BAC-*Pnky* mice to obtain a cohort of *Pnky*-KO (n = 15 XX and 18 XY) mice and littermate *Pnky*-KO;BAC-*Pnky* (n = 13 XX and 16 XY) mice. **b** *Pnky*-KO; BAC-*Pnky* XX mice exhibit decreased startle compared to its KO littermates (rank summary analysis overall effect *p = 0.0195 across the whole range of stimuli, followed by two-tailed, unpaired Welch's *t* tests at 0 dB: *p* = 0.6956, and 120 dB: **p* = 0.0044; two-tailed, Mann-Whitney test at 70 dB: *p* = 0.0907, 80 dB: *p* = 0.2999, 90 dB: *p* = 0.3162, 100 dB: *p* = 0.1696, 110 dB:

*p* = 0.0617) while for the XY animals there is no significant difference (rank summary analysis overall effect *p* = 0.8263) across the whole range of stimuli. **c** ASR reduction observed with BAC-*Pnky* in *Pnky*-KO XX mice (41.5%) is analogous to the amount of ASR increase observed with *Pnky*-KO vs. WT (55.45%) at 120 dB, while the XY mice did not have significant differences in ASR changes across genotypes. For the ASR test, n = 11 XX and 11 XY *Pnky*-WT, n = 12 XX and 14 XY *Pnky*-KO mice; and n = 15 XX and 18 XY *Pnky*-KO, n = 13 XX and 16 XY *Pnky*-KO;BAC-*Pnky* mice. Data is represented as mean ± SEM. Source data are provided as a Source Data file.

the XX-specific ASR phenotype of BAC-*Pnky* was not due to sex-specific differences in BAC-*Pnky* expression.

## Discussion

Animal behavior results from a complex interplay between the genome and the environment. Although many protein coding genes are known to be important to behavior by regulating brain development and/or function, the degree to which lncRNA genes underlie animal behavior is unclear. Relatively few lncRNAs have been studied in mice with genetic methods (e.g., gene disruptions and/or germline transgene insertions), and for only a handful of lncRNAs have the behavioral consequences of such genetic manipulations been evaluated[13,24]. In this study, we discovered that *Pnky*-KO mice have sex-specific behavioral

changes in fear context generalization and ASR. Furthermore, while XX *Pnky*-KO mice had increased ASR, the expression of *Pnky* from a BAC transgene decreased ASR in XX *Pnky*-KO mice, providing molecular-genetic evidence that *Pnky* can underlie animal behavior by functioning in *trans*.

The amplitude of the startle response is an important measure for evaluating affective deficits in patients with specific brain lesions (amygdala and frontal cortex), psychopathic tendencies, and a range of psychiatric conditions including panic disorder, bipolar disorder, and posttraumatic stress syndrome[36–38,40,41]. More specifically, increased startle amplitude is associated with greater anxiety levels in young female women[37]. While XX *Pnky*-KO mice had increased ASR amplitude, we did not observe differences in levels of anxiety as

assessed by the OFT and EPM. Furthermore, nest building behavior – an indicator of well-being in mice – was not altered in *Pnky*-KO mice of either sex. Thus, the change in ASR in XX mice lacking *Pnky* is not obviously related to a general increase in anxiety or change in the sense of well-being.

The ASR in rodents has been found to vary by biological sex, with male rats having increased ASR amplitude as compared to females[42,43]. In WT mice, we also observed higher ASR amplitude in XY versus XX mice. While *Pnky*-KO increased ASR amplitude in the XX mice, no difference in ASR was observed in XY mice across a wide range of acoustic intensities. Given that ASR is not modulated by the estrous cycle[42], we suggest that this important aspect of biological sex differences does not underlie the sex-specific role of *Pnky* in ASR.

Genetic manipulation such as deletion of the lncRNA locus combined with lncRNA expression from an independent transgene is a robust way to distinguish *cis* vs *trans* mechanisms[25,27]. The reversal of a lncRNA-KO phenotype by lncRNA expression in *trans* from a BAC establishes that the lncRNA can function in *trans*. We have previously demonstrated that expression of *Pnky* RNA from a BAC transgene in the *Pnky*-KO background can reverse the molecular, cellular and transcriptomic phenotypes of *Pnky* deletion[22].

In this study, while ASR amplitude was increased in XX *Pnky*-KO mice, BAC-*Pnky* reduced the ASR amplitude in XX *Pnky*-KO mice, which can be interpreted as a reversal of this *Pnky*-KO behavioral phenotype. Because we were unable to conduct the scale of animal husbandry required to generate the necessary numbers of all experimental groups to assess rescue (*i.e., Pnky*-WT, *Pnky*-WT;BAC-*Pnky*; *Pnky*-KO, and *Pnky*-KO;BAC-*Pnky*), we are unable to fully demonstrate "rescue" of the *Pnky*-KO phenotypes. However, we find it compelling that the reduction in ASR amplitude by BAC-*Pnky* was not observed in XY mice, indicating that this behavioral effect was specific and not reflective of a more general effect on ASR by the BAC-*Pnky* transgene.

In contrast to the ASR phenotype, fear context generalization in *Pnky*-KO mice was not changed by BAC-*Pnky* in either sex. Both fear context generalization and ASR behaviors are governed by activity and plasticity in different inter-connected brain regions such as the cortex, hippocampus, amygdala and the thalamus[38,44]. Based on RNA FISH studies, across these brain regions, *Pnky* was expressed at variably lower levels in *Pnky*-KO;BAC-*Pnky* as compared to WT mice. Thus, we speculate that certain behaviors require more precise regulation of *Pnky* expression for rescue in our assays.

Despite the lack of rescue by BAC-*Pnky*, the sex-specific behavioral change in fear generalization related to *Pnky*-KO are interesting to consider in the context of studies linking aberrant lncRNA expression and psychiatric diseases. Abnormal fear context generalization and recall is associated with anxiety, depression and many other mental health conditions[45]. Interestingly, in a small study of human blood samples, *PNKY* was found to be a strong molecular biomarker of treatment-resistant schizophrenia[46]. Overall, our finding that *Pnky*-KO mice are behaviorally grossly normal but exhibit phenotypes suggestive of affective disorders support the broader notion that lncRNAs in the brain serve to "fine-tune" the development and/or functions that underlie more complex behaviors.

In this study, we focused on behavioral studies and have not associated results with potential anatomic differences or alterations in neural cell function. Based on the literature, much of which is correlative, the underpinnings of the observed functional differences are very interesting but numerous. For instance, differences in ASR could relate to differences in amygdala or orbitofrontal cortex anatomy and/ or function, as these areas provide afferent modulation to the brainstem ASR circuits[38]. Previously, we found differences in cortical development in *Pnky*-KO mice[22], but connecting these anatomic differences with the observed behavioral phenotypes would be highly speculative. Furthermore, *Pnky* is also expressed in the adult mouse

brain, and it is therefore possible that this lncRNA modulates adult neuronal function. Currently, we hypothesize that specific anatomic differences in cell type and/or connectivity across multiple brain regions underlie the observed behavioral phenotypes, which warrants a comprehensive investigation.

Functional studies linking lncRNAs with animal behaviors comprise an emerging field of research that is important to our understanding of a wide range of neurological diseases including cognitive and psychiatric disorders. Because the breadth of potential mechanisms of lncRNAs is fundamentally wider (*cis* and *trans*) than most protein coding or microRNA genes, the strategies used to study lncRNA function may need to be tailored to each lncRNA[25,27,47]. The study of animal behavior is also experimentally complex, with results being sensitive to a large number of potential variables, and genetic tools are a powerful means to reducing the variability of lncRNA perturbation. Moving forward, additional in vivo studies of lncRNA function in animal behavior will become a crucial foundation for understanding how specific lncRNAs can underlie important disorders of the human mind.

## Methods
### Animals
All animal related research protocols comply with the relevant ethical regulations approved by the University of California, San Francisco Institutional Animal Care and Use Committee. C57BL/6 J mice used in this study were maintained in the University of California, San Francisco Laboratory Animal Resource Center under approved protocol (AN195649-01C). Mice were group-housed in cages with sterile bedding and *ad libitum* food and water. Standard conditions of 12-h dark/ light cycle, humidity (30–70%) and temperature (20–26 °C) were maintained. Mice of both sexes were used for all experiments and were analyzed at ages 3–6 months. All test results were analyzed relative to littermates. Estrous phase was not monitored for the "XX" genotype mice. Animals were ear-tagged, and the experimenter of the behavioral tests was blinded to the genotype of the animals. XX mice were run first and then XY to prevent their behavior being influenced by the exposure to the XY mice. Sexes were not inter-mixed. Mice were handled twice by the experimenter the week before the start of behavioral testing.

### Open field test
Open field activity is measured in a clear acrylic chamber (41 ×41 x 30 cm) with two 16 ×16 photobeam arrays that automatically detect horizontal and vertical movements, also known as the Flex-Field/Open Field Photobeam Activity System (San Diego Instruments, San Diego, CA). The acrylic chambers are located inside larger sound and light attenuating shells so that their spontaneous locomotor activity is not affected by external stimuli. Mice are allowed to habituate in the testing room under normal light for 60 min before testing. During testing, mice are placed in the center of the activity chamber and allowed to freely explore for 15 min.

### Elevated plus Maze test
The elevated plus maze consists of two open arms (without walls, 15" long X 2" wide) and two closed arms (with walls 6.5" tall), the intersection of the arms is 2" x 2" wide, and the entire maze is elevated 30.5" above the ground (Hamilton-Kinder, Poway, CA). Mice are first allowed to habituate in the testing room under dim light for 1 hr prior to the start of testing. During testing, mice are placed in the maze at the intersection of the open and closed arms and allowed to freely explore the maze for 10 min. The maze is cleaned with 70% alcohol between testing of each mouse. Main dependent measures are as follows 1) % Open arm time 2) Open arm/total distance 3) Closed arm distance 4) Total distance 5) Open arm entries 6) Closed arm entries 7) % Open arm time x minutes.

## Rotarod test

The training part of the test is used to introduce the mouse to the rotarod apparatus (Med Associates Inc., Vermont, USA) which is equipped with infrared beams that automatically detect when the mouse has fallen off the rotating rod. Each day of testing, the mice are allowed 1 hr to habituate to the procedure room before testing begins. On the first day, up to five mice of the same sex are simultaneously placed on the rotarod apparatus, and then it rotates at the constant speed of 16 RPM. The trial ends when the mouse falls off the rod or when 5 min has elapsed. The mice are tested on 3 individual trials with an inter-trial interval between 15–20 min.

The testing phase assesses their general motor learning ability or increase in performance over trials. On the second and third day of testing, five mice of the same sex are simultaneously placed on the rotarod apparatus with the rod rotating at an accelerated speed, from 4 RPM to 40 RPM. The rotation speed increases by 7.2 RPM every minute. The trial ends when the mouse falls off the rod or when 5 min has elapsed. The inter-trial interval is between 15–20 min. For the two testing days, the animals get a total of 6 trials per day separated into two 3-trial sessions with inter-session interval of ~3 h.

## Nesting

A standard mouse cage (10" x 7" x 6.5") is filled with ~2 cm of paper chip bedding and a single nestlet (5 cm square of pressed cotton batting) is placed in the center of the cage. Each mouse is single housed for the 24 hr testing period and the quality of their nest is scored at 2, 6, and 24 hrs after introduction of the nestlet. Nests are assigned a score from 0 to 7 at each time point. Scoring: 0 = nestlet untouched, 1 = less than 10% of the nestlet is shredded, 2 = 10–50% of the nestlet is shredded but there is no shape to the nest, 3 = 10–50% of the nestlet is shredded and there is shape to the nest, 4 = 50–90% of the nestlet is shredded but there is no shape to the nest, 5 = 50–90% of the nestlet is shredded and there is shape to the nest, 6 = Over 90% of the nestlet is shredded but the nest is flat, 7 = Over 90% of the nestlet is shredded and the nest has walls that are as tall as the mouse on at least 50% of its sides.

## 2-trial social approach test

Mice are brought into testing room and given one hour to acclimate to the room prior to testing. The social arenas are constructed from white acrylic. Dividers are clear acrylic with arch-shaped entrances at the center. For 2-chamber, arenas are divided into two 30 W x 40D x 23H cm chambers. The social arenas are housed inside sound attenuating shells measuring 80 W x 67D x 64H cm and equipped with two 0.5-amp lights and an exhaust fan. Social & non-Social enclosures are circular wire cups 10 cm in diameter and 13 cm tall. The spacing of the wire allows for olfactory and visual access between the stimulus and experimental mice. All stages of the experiment are videotaped from above using cameras mounted on the top of the sound attenuating shells. Mice are allowed 1 hr to acclimate to the procedure room before testing begins. The habituation phase and testing phase occurs back-to-back. During habituation the mouse is allowed to explore the entire arena with empty enclosures for ten minutes. During social approach trial, the mouse is allowed to freely explore the entire arena for another 10 min, but the enclosures now contain an inanimate mouse on one side and a live social stimulus mouse (non-transgenic, age, strain and sex matched, noncage/littermate) on opposite side. The experimental mouse is blocked in the nonsocial chamber before introducing the stimulus mouse and inanimate mouse into their respective enclosures. The location of the stimulus mouse is counterbalanced across genotypes. Videos are analyzed using CleverSys TopScan. The time and bouts spent in each chamber, in the proximity zone (5 cm area perimeter around enclosure), and "sniffing" (1.5 cm area perimeter around enclosure) is recorded. Difference scores (Social minus Non-Social) for bouts and time are calculated for the chamber side and interaction/proximity zones. Ratio scores (Social divided by Non-Social) for bouts and time are calculated for the chamber side and interaction/proximity zones.

## Object-context congruence test

Mice are moved into the testing room for 1 hr to acclimate under normal lighting conditions. Experiment is carried out inside sound attenuating shells equipped with two 5 W houselights, and an overhead video camera. Mice are randomly distributed into two groups: Context A first or Context B first for Trial 1, and reverse order for Trial 2. For the test phase, all the animals were run in the context that they had seen most recently (training trial 2) but with an incongruent object from the other context. Context A Chamber: plain white, Objects: white rectangular blocks, Cleaning Agent: 70% EtOH. Context B Chamber: checkered walls, Objects: 50 mL Erlenmeyer flasks with tented tops (filled with colored paper), Cleaning Agent: 1% Acetic Acid. Each trial is videotaped from above and videos are analyzed with post-acquisition software, CleverSys TopScan, to measure the time spent sniffing the objects within each Context (<1.5 cm distance around objects.). The objects are either cleaned with 70% EtOH or 1% Acetic Acid (dependent on Context) in between each mouse. Mice are tested in 1 day using three 10-min trials: 1) Trial 1 (Object-Context A or B pairing) for 10 min exploration then home cage for 30 min. 2) Trial 2 (Object-Context B or A pairing) for 10 min exploration then home cage for 4 hrs. 3) Object-Context Congruence Test in the most recent context with one congruent and one incongruent 10 min exploration then home cage.

## Cued fear conditioning and recall

Testing is done in the fear conditioning chamber 9.5" L x 12" W x 8.5" H (Med Associates) that sits inside a sound attenuating shell 25" L x 29.5' W x 14" H. The Med Associates VideoFreeze program is used for tracking. Mice are not habituated to the testing room on the day of testing and are brought into the testing room immediately before the trial begins.

Training (Day 1): Mice are placed using nitrile gloves in the fear conditioning apparatus with light set at 3, fan on, tray is sprayed with Windex, and the floor is an even-barred metal grid.

Test starts with a 5 min baseline period to measure baseline freezing activity. Then four-30 s 80 dB tones that co-terminate with a 2-s, 0.45 mA footshock are presented, separated by 120-s inter-cue interval (ICI) during which freezing is monitored. A 120 s ICI follows the last footshock. Freezing is monitored, with motion index threshold set to 18. Fear context test (Day 2): The context test takes place 24 h after Training. Setup is the exact same as training but a different scent is used. Mice are placed in the fear conditioning apparatus for 10 min. No cues or shocks are presented. Freezing is monitored. Cued fear recall Test (Day 3): takes place 24 h after the context test. The context is altered: A black insert covers the metal grid floor, and a black A frame is added. Chamber light set at 1, fan is off, and room lights are dimmed. Mice are placed with latex gloves into a different chamber from the training and context test phases for 15 min. After a 5 min context generalization period, four tones are delivered as described in Training. No shock is presented.

## Acoustic startle response

Testing is performed in a small, isolated chamber inside a sound-attenuating cubicle, free from external movement and noise (Hamilton-Kinder, Poway, CA). On the day of testing, an individual cage of group-housed mice or two single-housed mouse cages is/are transferred to the ante room just outside and prior to testing. The startle sensor plate is calibrated at the start of each testing day. The mice are placed into the restraining chamber and given 5 min to acclimate inside that restraining chamber before stimulus presentations begin. The restraining chamber is cleaned with 70% ethanol in between individual mice. Habituation: Mice are given 5 min to acclimate to the

restraining chamber and 64 dB background noise before acoustic stimulus testing begins. After 5 min, mice are exposed to a series of acoustic pulses at varying intensities for approximately 15 min at random intervals. Trials with no auditory stimulus are also included to provide a measure of baseline startle activity. The test is comprised of a total of 70 trials that are randomly selected from the following list: 10 trials each of 40 ms @120 dB, 40 ms @110 dB, 40 ms @100 dB, 40 ms @90 dB, 40 ms @80 dB, 40 ms @70 dB and no stimulus trials. The interstimulus interval (ISI) is variable with a mean of 15 s, and a range of 8–22 s. Average and Maximum amplitudes (N) of pulses are measured for each mouse.

## Pre-pulse inhibition
Testing is performed in a small, isolated restraining chamber placed inside a sound attenuating cubicle, free from external movement and noise (Hamilton-Kinder, Poway, CA). The startle sensor plate is calibrated at the start of each testing day. Habituation: Mice are given 5 min to acclimate to the restraining chamber and 64 dB background noise before acoustic stimulus testing begins. After 5 min, mice are exposed to a series of acoustic startle stimuli for 20 min in which some stimuli are preceded by a weaker acoustic stimulus (pre-pulse) at random intervals. Trials with no auditory stimulus are also included to provide a measure of baseline activity. The test is comprised of a total of 80 trials randomly selected from the following list: 24 trials of 40 ms@120 dB, and 14 trials each of the 4 dB pp 40 ms @120 dB, 15 dB pp 40 ms @120 dB, 26 dB pp 40 ms@120 dB, and no stimulus trials. The interstimulus interval (ISI) is variable with a mean of 15 s, and a range of 8–22 s. Average and Maximum amplitude of startle responses to each acoustic stimulus (measured in "N" calibrated units), as well as those with preceding pre-pulses, are recorded for each animal.

## Genotyping
All animals were genotyped by PCR for Pnky⁺, and Pnky⁻ alleles using the following primers: Pnky GT F: TAAGCTCAAACTCCGGTCCCGGGA, Pnky GT R1:TCAGGGACAAAGAACCAAAACGAGC, Pnky GT R2:AATGC TCCCTCTGAGCCTCAATT

Reaction products: 120 bp (*Pnky*-KO), 221 bp (*Pnky*-WT). Since BAC-*Pnky* contains unaltered *Pnky*, this will produce the 221-bp product, even in the absence of endogenous Pnky in the *Pnky*-KO;BAC-*Pnky* samples.

Primers for BAC-*Pnky*: BAC_F: CACCTGCTACCTGATATAGG, BAC_R: CCTGCTACCTGATATAGG. Reaction product: 416 bp (contains BAC-*Pnky*). No amplification in the absence of BAC-*Pnky*.

## Tissue dissection and RNA extraction
The brains of adult mice were dissected out of the skull after anaesthetizing the animals using Avertin. Microdissections were performed in ice-cold TransnetYX Hibernate media to collect the cortex, hippocampal region and amygdalar region in TRIzol™ reagent (Thermo-Fisher Scientific), homogenized using the pipette tip and the supernatant was collected for RNA extraction using Direct-zol RNA Purification Microprep Kits (Zymo Research). DNase digestion was performed as suggested.

## Quantitative reverse transcription PCR (RT-qPCR)
cDNA synthesis was performed using the Transcriptor first Strand cDNA synthesis kit (Roche) with oligodT and random hexamer primers. RT-qPCR was performed using SYBR Green (Roche) on a Light-Cycler 480 II (Roche). Relative gene expression was calculated using the ΔΔCt method, using GAPDH as a housekeeping gene for differential gene expression analyses. All RT-qPCR assays were performed using technical triplicate wells. qPCR primers used were as follows: Pnky_F: GGACATCTCCTTTCTCCGCC, Pnky_R: CACCAAGTGCTTTCTC AGCC, GAPDH_F: GGGAAATTCAACGGCACAGT, GAPDH_R:AGATG GTGATGGGCTTCCC.

## Tissue preparation
Transcardiac perfusion was performed on postnatal animals with phosphate-buffered saline (PBS) followed by 4% PFA. The brains were then dissected out of the skull and additionally fixed in 4% PFA O/N at 4 °C. All brain specimens were rinsed in PBS and then cryoprotected with 30% sucrose in PBS. Cryoprotected brains were then equilibrated in a 1:1 mixture of 30% sucrose and Tissue-Tek Optimal Cutting Temperature (OCT) (Thermo Fisher Scientific) for 1 hr at 4 °C, then frozen in a fresh batch of the same mixture using dry ice. Frozen brain blocks were equilibrated in the cryostat at −23 °C for at least 3 hrs before sectioning. Coronal sections with a thickness of 12 µm were collected on Superfrost Plus Microscope Slides (Thermo Fisher Scientific) and stored at −80 °C.

## RNA in situ hybridization
ISH was performed on tissue sections (prepared as described above) using the RNAscope Fluorescent Multiplex V2 kit (ACD Biosciences). Probes targeting the mouse *Pnky* transcript were used (RNAscope Probe-Mm-Pnky, Cat. # 405551). The slides were baked at 60 °C for 45 min. Then rinsed the slides in PBS for 10 min at RT with rotation to remove sucrose/OCT. Postfixed with 4% PFA for 15 min at 4 °C, then rinsed in PBS for 5 min at RT. Instead of submerging slides in boiling Target Retrieval solution, added preheated Target Retrieval solution on top of horizontal slides and incubated for 5 min at RT in the slide box. This was followed by wash with distilled water 2X by moving slides up and down 3–5 times and then final wash with 100% ethanol for 1 min and then slides were air-dried. ImmEdge Hydrophobic Barrier PAP Pen was used to draw a barrier around tissue sections. Protease plus was added to the sections and incubated for 12 min followed by washes in distilled water. For ISH, the standard RNAscope Multiplex Fluorescent v2 assay protocol was followed with probe hybridization in HybEZ oven for 2.5 hrs at 40 °C. Following incubations were performed in the oven at 40 °C: AMP 1 – 30 min, AMP 2- 30 min, AMP 3- 15 min, HRP-C1 –15 min, TSA Vivid™ Fluorophore 570 (1:1000 in TSA buffer) – 30 min, HRP blocker – 15 min with washes with 1X wash buffer for 2 min at RT between each step. DAPI was added at RT and then slides were mounted with Prolong Antifade Mountant (Invitrogen).

## Microscopy and image analysis
Samples were imaged using Leica Stellaris 5 confocal microscope. For RNAscope samples 63X oil immersion objective was used. One coronal section (around Bregma −1.34 to 1.46 mm) hemisphere per sample (n = 2–6 mice per sex and genotype) was tile-scanned for cortex (12 tiles in the somatosensory cortex), hippocampal region (18 tiles in the dentate gyrus region) and amygdalar region (9 tiles) with optical sections. All image analyses were performed using Fiji and 4 non-overlapping fixed areas per sample were quantified in Fiji (Release 2.15.1) for RNAscope punctae using the Cell-counter function.

## Figure preparation
Graphs were generated using GraphPad Prism (10.0.3) and figures using Adobe Illustrator (25.4.1).

## Statistics and reproducibility
No statistical method was used to predetermine sample size. No data were excluded from the analyses. The experiments were not randomized. The investigators were blinded to the allocation of genotypes during behavioral experiments and outcome assessment. The statistical details of each experiment can be found in the relevant figure legends and was performed using GraphPad Prism (10.0.3) and R (v 4.0.4).

## Reporting summary
Further information on research design is available in the Nature Portfolio Reporting Summary linked to this article.

## Data availability

Behavioral tests data generated during this study are available in the Supplementary Data 1. Processed data generated in this study are provided in the Source data file and Supplementary Information. Source data are provided with this paper.

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

## Acknowledgements

The authors thank members of the Lim lab for helpful discussions. We thank Sandra Chang (Lim Lab, UCSF) for work with animal husbandry and administrative expertise. We thank Iris Lo, Julia Holtzman, and Jessica Speckart for contributions to the behavioral analyses within the Gladstone Institutes Behavioral Core Facility. This work was supported by NIH award 1R01NS124881 and Veterans Affairs 5I01 BX000252 to D.A.L.

## Author contributions

R.E.A. and D.A.L. conceptualized the study and designed experiments with guidance from J.S. P.S., R.E.A., J.S. and D.A.L. analyzed data, and P.S. and D.A.L. prepared the figures. P.S., S.J.H., E.G. and H.C. performed and analyzed experiments. P.S. and D.A.L. wrote the manuscript. All authors reviewed and edited the manuscript.

## Competing interests

The authors declare no competing interests.
