## [Peer Review File · Nature Communications]

Sex-specific role for the long noncoding RNA Pnky in mouse behaviorREVIEWER COMMENTS

Reviewer #1 (Remarks to the Author):

This study by Saha et. al examines the role of the long noncoding RNA (lncRNA) Pnky in several mouse behaviors in female and male mice. The authors argue that the novelty and impact of the study come from being able to demonstrate that lncRNA can function in trans and be associated with specific behaviors. The behavioral analysis is in large part very well done. The study shows that male but not female Pnky-KO mice exhibit lower baseline freezing and freezing to the tone in cued fear conditioning relative to same- sex Wildtype (WT) mice and find that female but not male Pnky-KO mice exhibit increased acoustic startle response (ASR) and decrease prepulse inhibition (PPI) relative to same- sex WT mice. A rescue experiment is carried out to express Pnky from a bacterial artificial chromosome (BAC) transgene which is found to reverse altered behavior due to the knockout in ASR but no other behaviors. Overall, this study is interesting and important as it pushes the envelope on assessing the impact of lncRNAs in behavior, however, a number of concerns are noted including interpretation of behavioral results and validation of manipulations. These, and other concerns are listed in more detail below.

1) Authors report in Figure 3 that male Pnky-KO mice have deficits in cued fear recall. The interpretation of lower freezing in Pnky-KO males vs WT males is problematic however given that that baseline levels of freezing prior to tone onset are also lower in KO mice, vs WT. Authors should consider reporting data with the baseline freezing subtracted to account for this. If baseline freezing is subtracted, does a difference still exist between the KO and WT groups or is this due to initial baseline freezing differences? If a difference does not exist, authors should consider reporting and discussing data as baseline differences associated with context, instead of a deficit in cued fear test. To further this point, Figure 3 Supplement 3C, KO male data looks similar to WT data in Figure 3. Authors should also report shock reactivity data in supplemental to account for differences in genotype or sex.

2) There is no validation of loss of Pnky expression in the Pnky-KO mice. That is in another study, but would be good to confirm in the animals used in this study. Additionally, understanding baseline expression of Pnky in WT mice of both sexes and in brain regions required for mouse behaviors assessed are important bases to cover.

3) It is important to show expression of Pnky from the BAC in the brain regions associated with ASR, PPI, etc. This is a necessary control, especially with regard to male mice, which had no change in any behavior from the BAC. That could be explained by lack of expression from the BAC, which although unanticipated, could have happened. Authors state that it may be possible that BAC-Pnky mice do not fully recapitulate the levels of Pnky expression in WT mice. Thus, expression data will help with interpretation of the behavioral observations.

4) The extent of developmental abnormalities in Pnky KO mice is not clear. For example, the authors mention the cortical development abnormalities, which may be key to interpretation of the behavior. Are there abnormalities in the amygdala, hippocampus, etc? Related to this is that if the BAC does truly rescue the impairments (but see issues related to interpretation of those experiments) then a deeper discussion about the role of Pnky in development versus adult brain function would strengthen the manuscript.

5) A main conclusion of the study is that Pnky is functioning in trans. This appears to be based on a previous study showing that Pnky does not function in cis (Ramos 2015; Andersen 2019). However, over-expression by the BAC may lead to cis effects. The possible cis regulation by Pnky must be examined in this study in the brain regions involved in the behaviors assessed to make the conclusion that it is purely a trans-dependent mechanism.

6) As the main phenotype in males is cued fear recall, it is important to demonstrate that there was not a testing order effect (context then cued) by running a cued fear test first.

7) There is a performance difference in the Pnky KO male mice with regard to freezing behavior in the experiments shown in Figure 3 and Supplemental Figure 5. In the supplemental data, the Pnky KO mice are performing just like the WT mice in Figure 3. This raises a concern about whether these mice truly have a cued fear recall problem. The data is difficult to interpret without that control.

8) In ASR there appears to be differences from experiment to experiment in startle response, which makes comparing the BAC data in Figure 5 to the KO/WT data in Figure 4 a bit problematic. The data is shown as the KO being set to 1.0 in Figure 5C to resolve that difference. But the issue is that WT animals were not run (or at least data not shown) in the BAC experiment. Thus, comparing these experiments as shown in Figure 5C is problematic. The authors write in the discussion that they were not able to generate the WT control for that experiment, but it is still a necessary control.

9) There is an overstatement about Pnky being required for normal PPI in female mice as it was only the lowest 4 dB pre-pulse that showed effect and it seems driven by four animals. Thus, not very convincing.

Minor concerns:

-Pg. 5 lines 200-201 please include references.

-Please state whether estrous phase was monitored in methods.

- Please state whether mice were group or single housed. Also feeding and housing conditions are missing.
- Which order were mice run in for behavioral tests? Were sexes intermixed or was one sex run before the other? Encoding this information in the methods would be important to note as scent of a male or male urine can cause a female to go into proestrus which may impact behavior.
- Were mice handled at all by experimenters prior to behavior? Please include in methods.
- Was the social-stimulus mouse sex-matched in the 2-trial social approach task? Please include in methods.
- Please include motion index and threshold parameters measuring freezing in the VideoFreeze software. It is also not stated whether the same or a different scent was used in the cue test context.

Reviewer #2 (Remarks to the Author):

The study entitled "Sex-specific role for the long noncoding RNA Pnky in mouse behavior" continues to explore the role of the lncRNA Pnky. In a previous study the Lim lab, quite heroically, demonstrated that Pnky is a trans-acting lncRNA that plays critical roles NSC development and ultimately in cortical development. Here they extend these studies and mouse models to understand the role of Pnky in animal behavior - a relatively unexplored area of research. Specifically, they explore the role of Pnky in nest building, social interactions, cognitive behaviors, fear conditioning (in multiple phases) and acoustic startle.

Interestingly, the authors find that many behaviors are not different between Pnky KO and WT in both XX and XY genotypes. Yet, there are genotype-based behavioral differences in the third phase of fear conditioning and acoustic startle responses. Specifically, the authors find that XY Pnky, but not XX, have less fear conditioning to cued fear stimulation. Moreover, the acoustic startle response (ASR) is increased in XX genotypes. This finding was further investigated using repulse inhibition (PPI) approaches. Consistent with the previous ASR phenotype it was found that XX genotypes have reduced PPI.

Then the authors test whether trans-gene (from a BAC) can rescue the observed ASR and PPI phenotypes observed in XX genotypes. Moreover, the authors tested if transgenic expression could rescue the fear conditioning response observed in XY genotypes. These studies found that XY fear conditioning and XX PPI defects could not be rescued. In contrast the authors observed a reversal of the XX, ASR phenotype.

The negative results point to the importance of the positive result that a lncRNA can rescue behavioral issues in knockout background.

Collectively, this is a well-written, transparent and informative study breaking the ground for future research on lncRNAs in animal behavior. It is this reviewer's opinion that this study will be of great interest to the general readership of Nature Communications. Below are a few comments and curiosity about this study.

Curiosity

1) What happens in this same set of behavioral tests if Pnky is induced in a WT background? It could be that in some diseases there is a gain of function of Pnky later in development or adulthood. This would be confounded by early brain development defects observed in their previous study. But could Pnky be induced for a few days in adulthood (or a time point the authors feel is relevant) and then test for these behaviors. It may be beyond the scope of this current study but would provide an equal and opposite insight into the role of lncRNAs in animal behavior (from a GOF perspective).

Minor comments

1) It would be helpful to include the effect size (e.g, fold-change) and P-Value in text next to each test conducted.

2) It may be more amenable to "gender neutrality" to describe "male" as XY genotype and "female" as XX genotype. It is a growing concern amongst younger generations that could provide more "neutral" results outside the context of "gender". Having said that, this reviewer did not find it an issue and just thought to have the important results not be distracted as a "gender" based phenotype.

RESPONSE TO REVIEWERS

Thank you for your helpful and insightful comments regarding our manuscript, “**Sex-specific role for the long noncoding RNA *Pnky* in mouse behavior**,” We have performed data analyses and new experiments to support our findings and enhance our conclusions. With this new data and changes to the text, we believe that the paper has been greatly improved.

Below, we have responded to all comments, and each reviewer is addressed in full separately. We hope with the additional clarity that these experiments provide, you will find our manuscript suitable for publication in Nature Communications.

All significant changes to the text in the manuscript are in blue font.

The new data can be found in the following figures:

Figure 3 (additional supplementary Fig 6C)

Figure 5 (modified Fig. 5C)

Supplementary Fig. 1

Supplementary Fig. 5

Supplementary Fig. 10

Supplementary Fig. 12

Thank you again for your time and consideration.

REVIEWER COMMENTS

Reviewer #1 (Remarks to the Author):

This study by Saha et. al examines the role of the long noncoding RNA (lncRNA) *Pnky* in several mouse behaviors in female and male mice. The authors argue that the novelty and impact of the study come from being able to demonstrate that lncRNA can function in trans and be associated with specific behaviors. The behavioral analysis is in large part very well done. The study shows that male but not female *Pnky*-KO mice exhibit lower baseline freezing and freezing to the tone in cued fear conditioning relative to same- sex Wildtype (WT) mice and find that female but not male *Pnky*-KO mice exhibit increased acoustic startle response (ASR) and decrease prepulse inhibition (PPI) relative to same- sex WT mice. A rescue experiment is carried out to express *Pnky* from a bacterial artificial chromosome (BAC) transgene which is found to reverse altered behavior due to the knockout in ASR but no other behaviors. Overall, this study is interesting and important as it pushes the envelope on assessing the impact of lncRNAs in behavior, however, a number of concerns are noted including interpretation of behavioral results and validation of manipulations. These, and other concerns are listed in more detail below.

We thank the reviewer for their positive evaluation of the manuscript and helpful comments regarding the interpretation of behavioral results. The comments were instrumental in guiding our revisions and the addition of new data.

1) Authors report in Figure 3 that male *Pnky*-KO mice have deficits in cued fear recall. The

interpretation of lower freezing in *Pnky*-KO males vs WT males is problematic however given that that baseline levels of freezing prior to tone onset are also lower in KO mice, vs WT. Authors should consider reporting data with the baseline freezing subtracted to account for this. If baseline freezing is subtracted, does a difference still exist between the KO and WT groups or is this due to initial baseline freezing differences? If a difference does not exist, authors should consider reporting and discussing data as baseline differences associated with context, instead of a deficit in cued fear test. To further this point, Figure 3 Supplement 3C, KO male data looks similar to WT data in Figure 3. Authors should also report shock reactivity data in supplemental to account for differences in genotype or sex.

As suggested, we analyzed the freezing response in the cued fear recall test taking into consideration the baseline freezing differences between WT and KO animals (**new Supplementary Fig. 6c**). Given that the freezing behavior was reduced during context generalization in male *Pnky*-KO mice, and differences during tone presentation and the inter-cue interval could be attributed to this, we agree with the reviewer that the most conservative interpretation of these data is that there is a decrease in fear context generalization in male *Pnky*-KO mice: “Given that the freezing behavior was reduced during context generalization in XY (male) *Pnky*-KO mice, and differences during tone presentation and the ICI could be attributed to this (Supplementary Fig. 6c), the most conservative interpretation of these data is that there is a decrease in context generalization in XY (male) *Pnky*-KO mice.”

As requested, in **new Supplementary Fig. 5 and 10**, we have added the shock reactivity data (Motion Index Test) and hind paw withdrawal latency data. There are no differences across the different genotypes and sex.

2) There is no validation of loss of *Pnky* expression in the *Pnky*-KO mice. That is in another study, but would be good to confirm in the animals used in this study. Additionally, understanding baseline expression of *Pnky* in WT mice of both sexes and in brain regions required for mouse behaviors assessed are important bases to cover.

We have performed RT-qPCR to confirm the absence of *Pnky* transcripts in the cortex, hippocampus and amygdala of *Pnky*-KO mice (**Supplementary Fig. 1**, reproduced below).

Supplementary Fig. 1: *Pnky* expression levels in **a)** cortex, **b)** hippocampus and **c)** amygdala by qRT-PCR relative to *Pnky*^{+/+} mean for respective sex. Quantification: mean \pm SEM of biological replicates. Statistical analyses: unpaired t test (two-tailed). ns, not significant, * $p < 0.05$, ** $p < 0.001$

We also have performed RNA FISH for *Pnky* RNA in a subset of animals used for the behavioral testing (**Supplementary Fig. 12**, quantification panels reproduced below).

We find that the fluorescent signal with *Pnky* probes in *Pnky*-KO animals was significantly less than that of controls; given the absence of transcripts detected by RT-qPCR in Supplementary Fig. 1 and previous studies (Andersen *et al.*, *Dev Cell* 2019), and the fact that the *Pnky*-KO mice are a germline *Pnky* genomic deletion, the minor amount of *Pnky* FISH signal in the *Pnky*-KO mice is most likely non-specific background signal.

Supplementary Fig. 12: Quantification of RNAScope *Pnky* signal presented as Number of *Pnky* foci per nuclei in **b**) cortex (100-120 nuclei) **d**) hippocampus (100-150 nuclei) and **f**) amygdala (20-40 nuclei) were counted across 3-4 non-overlapping fixed areas per animal (n= 2-6 animals per sex and genotype). unpaired t -test (two-tailed) was used and data is represented as mean \pm SEM. ns, not significant. * $p < 0.05$, ** $p < 0.01$, *** $p < 0.001$, **** $p < 0.0001$.

3) It is important to show expression of *Pnky* from the BAC in the brain regions associated with ASR, PPI, etc. This is a necessary control, especially with regard to male mice, which had no change in any behavior from the BAC. That could be explained by lack of expression from the BAC, which although unanticipated, could have happened. Authors state that it may be possible that BAC-*Pnky* mice do not fully recapitulate the levels of *Pnky* expression in WT mice. Thus, expression data will help with interpretation of the behavioral observations.

This comment is also addressed by the *Pnky* RNA FISH experiment above. As expected, similar to previous studies (Andersen *et al.*, 2019), *Pnky* expression was deficient in *Pnky*-KO brain regions, and *Pnky* levels in *Pnky*-KO;BAC-*Pnky* mice were 32-55% of control across the different brain regions (**Supplementary Fig. 12**). Expression of *Pnky* from the BAC transgene was detected in both XY and XX mice at similar levels, thus, the XX-specific ASR phenotype of BAC-*Pnky* was not due to sex-specific differences in BAC-*Pnky* expression. We agree with the

reviewer that these data help interpret the behavioral observations, and this is now also included in the Discussion.

4) The extent of developmental abnormalities in *Pnky* KO mice is not clear. For example, the authors mention the cortical development abnormalities, which may be key to interpretation of the behavior. Are there abnormalities in the amygdala, hippocampus, etc? Related to this is that if the BAC does truly rescue the impairments (but see issues related to interpretation of those experiments) then a deeper discussion about the role of *Pnky* in development versus adult brain function would strengthen the manuscript.

In an attempt to address the developmental questions raised, we analyzed brain sections from the cohorts used for the behavioral studies. Given that we have previously reported defects in cortical lamination in prenatal and early postnatal *Pnky* KO animals, we first performed IHC for CUX1 (upper layer neuron). With the subset of brains that were preserved from the behavioral studies (cohort 1 WT: 4 males, 4 females; cohort 1 KO: 3 males, 2 females; cohort 2 KO: 4 males, 4 females; cohort 2: 4 males, 4 females; total n= 29 animals) we observed a decrease in the CUX1+ cells with *Pnky*-KO, and there was an increase in CUX1+ cells in *Pnky*-KO;BAC-*Pnky* mice as compared to *Pnky*-KO (**Reviewer Figure**, below). However, this difference was only statistically significant if we pooled the 2 cohorts together. We do not think that it is experimentally appropriate to pool the two cohorts. Thus, our interpretation is that this experiment is not sufficiently powered to detect the likely subtle developmental abnormalities of the *Pnky*-KO cortex.

Reviewer Fig : a) CUX1 IHC in the upper layers of the cortices of *Pnky*-WT, *Pnky*-KO (two cohorts) and *Pnky*-KO; BAC-*Pnky* animals (n= 2-4 mice per genotype). DAPI was used to label nuclei. Scale bar = 100 μ m. **b)** Quantification: mean \pm SEM of biological replicates.

Similarly, we examined the hippocampal and amygdalar regions and did not observe any obvious anatomical differences across the genotypes. We agree with the reviewer that a deeper discussion about the potential roles of *Pnky* in development versus adult brain function would be interesting, and we have added this to the Discussion.

5) A main conclusion of the study is that *Pnky* is functioning in trans. This appears to be based on a previous study showing that *Pnky* does not function in cis (Ramos 2015; Andersen 2019). However, over-expression by the BAC may lead to cis effects. The possible cis regulation by *Pnky* must be examined in this study in the brain regions involved in the behaviors assessed to make the conclusion that it is purely a trans-dependent mechanism.

We used genetic strategies to test the mechanistic concept of *trans* function. In the lncRNA field, a rigorous test of *trans* function is to express the lncRNA gene from a BAC transgene in the context of the lncRNA KO (reviewed in Kopp and Mendell, *Cell* 2018; Bassett et al., *eLife* 2014). In our 2015 Ramos et al paper, we did not find evidence of cis regulation (*Pnky* knockdown did not alter gene expression in a 1MB window around the *Pnky* locus). In the 2019 Andersen et al paper, we developed genetic methods (expression of *Pnky* from a BAC transgene in the context of *Pnky*-KO) to show that cellular and transcriptomic (including splicing) phenotypes are rescued by *Pnky* provided in *trans* (i.e., from an integrated BAC-*Pnky* transgene). In this paper, we extended this definition of *trans* function to our behavioral studies.

To address the reviewer's concern, we have slightly modified the text to ensure the most conservative interpretation of the genetic manipulations. For example, we have added the word "can" to the conclusion "... this lncRNA *can* underlie specific behavior by functioning in *trans*," which is supported by the BAC-*Pnky* transgene decreasing acoustic startle response (ASR) in the *Pnky*-KO mice.

Of note, we did not find evidence of strong over-expression of *Pnky* from the BAC-transgene (**Supplementary Fig. 12**) in the 2 cohorts of mice used in our behavioral studies. In fact, the expression of BAC-*Pnky* appears to be about 32-55% of control levels. Nevertheless, we do not believe that this alters the genetic distinction between *cis* and *trans* function.

6) As the main phenotype in males is cued fear recall, it is important to demonstrate that there was not a testing order effect (context then cued) by running a cued fear test first.

Based on the reviewer's suggestion in comment #1, we now report the male phenotype as a decrease in context generalization, not cued fear recall. While we appreciate that the reviewer here was suggesting a way in which we might be able to distinguish cued fear recall from context generalization, to maintain the cue salience (and allow the scoring for context generalization and discrimination), we did not perform the cued fear conditioning or recall test running a cued fear test first.

7) There is a performance difference in the *Pnky* KO male mice with regard to freezing behavior in the experiments shown in Figure 3 and Supplemental Figure 5. In the supplemental data, the *Pnky* KO mice are performing just like the WT mice in Figure 3. This raises a concern about

whether these mice truly have a cued fear recall problem. The data is difficult to interpret without that control.

We agree that there is a limitation to our ability to make comparisons across the two experimental cohorts (cohort 1 = WT vs. KO; cohort 2 = KO vs. KO+BAC-*Pnky*). These two cohorts were derived from different breeding pairs, and the behavioral tests were run as two separate experimental cohorts. Given these differences, we do not draw conclusions about the percentage of freezing behavior across the two different experimental cohorts. The reviewer here points out that for context generalization, in cohort 1, male *Pnky*-WT mice had 26.2% freezing behavior and *Pnky*-KO mice had 13.6% freezing behavior, whereas in cohort 2, *Pnky*-KO mice had about 20.1% freezing behavior. Of note, in the second cohort, the percentage of freezing behavior in male mice was generally higher in all assays as compared to the first cohort, further highlighting the cohort-dependent differences. Nonetheless, with the rigorous experimental design of cohort 1, we believe that our data strongly support the conclusion that freezing behavior in context generalization is decreased in male *Pnky*-KO mice as compared to male control.

8) In ASR there appears to be differences from experiment to experiment in startle response, which makes comparing the BAC data in Figure 5 to the KO/WT data in Figure 4 a bit problematic. The data is shown as the KO being set to 1.0 in Figure 5C to resolve that difference. But the issue is that WT animals were not run (or at least data not shown) in the BAC experiment. Thus, comparing these experiments as shown in Figure 5C is problematic. The authors write in the discussion that they were not able to generate the WT control for that experiment, but it is still a necessary control.

As with point 7 above, we agree with the reviewer that there is a limitation in our ability to make comparisons across experimental cohorts. In the previous Figure 5C, we had set startle magnitude to 1.0 for *Pnky*-KO mice, to “normalize” comparisons across the two cohorts.

To make these data more transparent, we now show the actual values of the startle response on the y-axis, which shows the

Figure 5c ASR reduction observed with BAC-*Pnky* in *Pnky*-KO XX mice (41.5%) is analogous to the amount of ASR increase observed with *Pnky*-KO vs. WT (55.45 %) at 120dB, while the XY mice did not have significant differences in ASR changes across genotypes.

differences across the two cohorts (**new Figure 5C**, shown above). Importantly, this new representation of the data does not change our interpretation of the results, that BAC-*Pnky* produces a decrease in ASR, which is the opposite phenotype as that observed with *Pnky*-KO (an increase in ASR).

With regard to both comments 7 and 8 above, we understand that being able to assess *Pnky*-WT, *Pnky*-KO and *Pnky*-KO;BAC-*Pnky* animals in the same experiment would have been ideal. Initially, when designing these experiments, we considered the crosses necessary to generate the “ideal” experiment. This would be *Pnky*+/- crossed to *Pnky*+/-;BAC-*Pnky*. The expected frequency of the *Pnky*+/, *Pnky*-KO, and *Pnky*-KO;BAC-*Pnky* from this cross would be 0.0625 to 0.125. The average litter size is 6. Thus, each litter would produce between 0.375 to 0.75 mice of the required *Pnky* genotype. With group sizes of ~30 mice (15 males and 15 females) required within an age-matched cohort, we’d need at least 80 mating pairs to successfully mate within the same month or so. Given that our success rate of mating is a bit less than 50% per mating pair, this would require around 160 breeding cages. This would require the generation of 160 *Pnky*+/- mice and 160 *Pnky*+/-;BAC-*Pnky* mice of breeding age within a relatively close time window. Thus, the generation of age-matched littermate WT controls for these behavioral studies is beyond the scope of this revision, and from a practical standpoint, we were unable to execute this design.

9) There is an overstatement about *Pnky* being required for normal PPI in female mice as it was only the lowest 4 dB pre-pulse that showed effect and it seems driven by four animals. Thus, not very convincing.

We have modified the results to interpret our results more conservatively: “Given that PPI was reduced only at low prepulse intensity, and that a high ASR can contribute to that observation, the most conservative interpretation of these results is that *Pnky*-KO increases the ASR in XX (female) mice.” However, we beg to differ that this effect is only driven by 4 animals, as the majority of *Pnky*-KO mice had startle inhibition lower than the mean of *Pnky*-WT mice. Nevertheless, we understand the reviewer’s sentiment and have de-emphasized this observation in the text.

Minor concerns:

-Pg. 5 lines 200-201 please include references.

We have included the reference.

-Please state whether estrous phase was monitored in methods.

We have included in the “Methods”, the estrous phase was not monitored.

-Please state whether mice were group or single housed. Also feeding and housing conditions are missing. –

We have included in the “Method”, the mice were group housed with *ad libitum* food and water.

-Which order were mice run in for behavioral tests? Were sexes intermixed or was one sex run

before the other? Encoding this information in the methods would be important to note as scent of a male or male urine can cause a female to go into proestrus which may impact behavior.-

Females were run first and then males to prevent their behavior being influenced by exposure to male or male scent. Sexes not intermixed.

-Were mice handled at all by experimenters prior to behavior? Please include in methods.

Mice were handled twice by the experimenters prior to behavioral testing.

-Was the social-stimulus mouse sex-matched in the 2-trial social approach task? Please include in methods-

Yes, the social stimulus mouse was sex-matched.

-Please include motion index and threshold parameters measuring freezing in the VideoFreeze software. It is also not stated whether the same or a different scent was used in the cue test context.

We have included the motion index data in the supplementary material. Threshold was set at 18. Different scent was used in the cue test context.

Reviewer #2 (Remarks to the Author):

The study entitled "Sex-specific role for the long noncoding RNA Pnky in mouse behavior" continues to explore the role of the lncRNA Pnky. In a previous study the Lim lab, quite heroically, demonstrated that Pnky is a trans-acting lncRNA that plays critical roles NSC development and ultimately in cortical development. Here they extend these studies and mouse models to understand the role of Pnky in animal behavior - a relatively unexplored area of research. Specifically, they explore the role of Pnky in nest building, social interactions, cognitive behaviors, fear conditioning (in multiple phases) and acoustic startle.

Interestingly, the authors find that many behaviors are not different between Pnky KO and WT in both XX and XY genotypes. Yet, there are genotype-based behavioral differences in the third phase of fear conditioning and acoustic startle responses. Specifically, the authors find that XY Pnky, but not XX, have less fear conditioning to cued fear stimulation. Moreover, the acoustic startle response (ASR) is increased in XX genotypes. This finding was further investigated using repulse inhibition (PPI) approaches. Consistent with the previous ASR phenotype it was found that XX genotypes have reduced PPI.

Then the authors test whether trans-gene (from a BAC) can rescue the observed ASR and PPI phenotypes observed in XX genotypes. Moreover, the authors tested if transgenic expression could rescue the fear conditioning response observed in XY genotypes. These studies found that XY fear conditioning and XX PPI defects could not be rescued. In contrast the authors observed a reversal of the XX, ASR phenotype. The negative results point to the importance of the positive result that a lncRNA can rescue behavioral issues in knockout background.

Collectively, this is a well-written, transparent and informative study breaking the ground for future research on lncRNAs in animal behavior. It is this reviewer's opinion that this study will be of great interest to the general readership of Nature Communications. Below are a few comments and curiosity about this study.

We thank the reviewer for the positive evaluation of the manuscript and helpful suggestions.

Curiosity

1) What happens in this same set of behavioral tests if *Pnky* is induced in a WT background? It could be that in some diseases that there is a gain of function of *Pnky* later in development or adulthood. This would be confounded by early brain development defects observed in their previous study. But could *Pnky* be induced for a few days in adulthood (or a time point the authors feel is relevant) and then test for these behaviors. It maybe beyond the scope of this current study but would provide an equal and opposite insight into the role of lncRNAs in animal behavior (from a GOF perspective).

We agree with the reviewer that studying the behavioral phenotypes in *Pnky* over-expression will be an interesting paradigm. However, we currently do not have reliable tools for inducing the expression of *Pnky* in adulthood. We agree that such experiments would be beyond the scope of the current study.

Minor comments

1) It would be helpful to include the effect size (e.g, fold-change) and P-Value in text next to each test conducted.

We have included the *p* values and percent changes in the text.

2) It maybe be more amenable to "gender neutrality" to describe "male" as XY genotype and "female" as XX genotype. It is a growing concern amongst younger generations that could provide more "neutral" results outside the context of "gender". Having said that, this reviewer did not find it an issue and just a thought to have the important results not be distracted as a "gender" based phenotype.

As suggested, we have modified "Females" and "males" to "XX" and "XY" genotype.

REVIEWERS' COMMENTS

Reviewer #1 (Remarks to the Author):

I appreciate the additional experiments, re-analysis, revised figures, and most importantly the adjusted conclusions based on the results given the limitations that have been raised. The authors have responded to everything raised and modified several conclusions in a manner that better reflects the caveats and limitations.

Reviewer #2 (Remarks to the Author):

The authors have addressed my suggestions.